# *Tristaenone A*: A New Anti-Inflammatory Compound Isolated from the Australian Indigenous Plant *Tristaniopsis laurina*

**DOI:** 10.3390/molecules27196592

**Published:** 2022-10-05

**Authors:** Shintu Mathew, Xian Zhou, Gerald Münch, Francis Bodkin, Matthew Wallis, Feng Li, Ritesh Raju

**Affiliations:** 1NICM Health Research Institute, Western Sydney University, Penrith, NSW 2751, Australia; 2Department of Pharmacology, Western Sydney University, Campbelltown Campus, Sydney, NSW 2560, Australia; 3School of Science, Western Sydney University, Penrith, NSW 2751, Australia

**Keywords:** Australian indigenous plant, hydroxycyclohexenone, flavonoids, anti-inflammatory, nitric oxide (NO), nuclear factor kappa B (NF-κB)

## Abstract

Inspired by ethnopharmacological knowledge, we conducted a bioassay-guided fractionation of the leaves of *Tristaniopsis laurina* which led to the discovery of a new anti-inflammatory compound, tristaenone A (**1**). The structure was elucidated by detailed spectroscopic data analysis, and the absolute configuration was established using X-ray crystallography analysis. Tristaenone A (**1**) suppressed LPS and IFN-γ-induced NO, TNF-α and IL-6 production in RAW 264.7 cells with IC_50_ values of 37.58 ± 2.45 μM, 80.6 ± 5.82 μM and 125.65 ± 0.34 μM, respectively. It also inhibited NF-κB nuclear translocation by 52.93 ± 14.14% at a concentration of 31.85 μM.

## 1. Introduction

Inflammation is a biological response to various stimuli such as pathogens, damaged cells, toxic compounds, pollutants or irradiation [1]. Prolonged exposure to these stimuli can lead to the progression of multiple inflammatory diseases including cancer, obesity and cardiovascular disorders [2]. Nonsteroidal anti-inflammatory drugs (NSAIDs) are among the commonly used drugs for the treatment of inflammation-related diseases. However, cardiovascular risks and gastrointestinal side effects from long-term use encourage the need for the discovery of new anti-inflammatory agents with minimal side effects [3,4].

Macrophages are essential immune cells that play a crucial role in the process of inflammation. Macrophages on activation produce large amounts of various pro-inflammatory mediators, such as nitric oxide (NO), prostaglandin-E2 (PGE2), pro-inflammatory cytokine (e.g., interleukin-6 (IL-6)) and tumor necrosis factor-alpha (TNF-α) [5]. Studies have reported that overproduction of NO has been involved in developing inflammation and in the progression of various inflammatory diseases [6]. Hence, the inhibition of excessive production of NO may have a therapeutical benefit in controlling inflammation.

Traditional remedies and natural products are an alternative to clinical drugs which have severe side effects, and they hold considerable promise in terms of identifying bioactive lead molecules and developing them into therapeutics to treat inflammatory illnesses [7,8]. Australian aboriginals have a long history of using medicinal plants to treat ailments such as cough, sore throat, wounds and skin infections [9,10].

Our ongoing investigations for the discovery of new anti-inflammatory agents made us focus on the leaves of *Tristaniopsis laurina* (Myrtaceae) (Peter G. Wilson and J.T Waterh) based on existing ethnopharmacological knowledge documented by Francis Bodkin on its use to heal sores and ulcers [11]. *T. laurina* is an evergreen Australian native tree that comes under the family Myrtaceae and typically grows to a height of 30 m. These trees are mainly distributed in south-eastern Queensland, eastern New South Wales and eastern Victoria [12].

Here, we describe the isolation, structure elucidation and anti-inflammatory activity of the new compound, tristaenone A (**1**) and two known flavonoids 8-desmethyleucalyptin (**2**) and eucalyptin (**3**).

## 2. Results and Discussion

Tristaenone A (**1**) was obtained as a colourless crystal. HRESI (+) MS analysis returned a molecular formula (C_24_H_24_O_5_Na) requiring thirteen double bond equivalents. The NMR (methanol-*d*_4_) data (Table 1) revealed resonances of two isolated monosubstituted aromatic systems, (C-1–C-7) reminiscent of a benzoate residue and a second system being a phenyl keto residue (C-1′–C-7′) (Figure 1). Key HMBC correlations from the aromatic resonances to H-3/7 (δ_H_ 8.09) to the ester carbonyl C-1 (δ_C_ 166.9) confirmed the benzoate fragment (Figure 1), while HMBC correlations from H-3′/7′ (δ_H_ 7.60) to the ketone C-1′ (δ_c_ 199.3), confirmed the phenyl ketone residue (Figure 1). An extension of the benzoate subunit was established through key HMBC correlations of the singlet oxy-methine H-5″ (δ_H_ 5.52/δ_c_ 80.2) to several carbons, C-1 (δ_C_ 166.9), C-3″ (197.2), C-4″/6″ (46.2), C-7″/9″ (22.7) and C-8″/10″ (25.8), (Figure 1). The presence of several HMBC correlations from the oxy-methine to several geminal methyls all bearing the same chemical resonance (δ_H_ 1.40, s) and integrating for 12 hydrogens suggested a chemically equivalent environment, with the four geminal methyl’s flanked between the oxymethine H-5″ (Figure 1). The presence of the two substructures A and B generated so far accounted for a total of 11 DBE’s, short of two more DBE’s. Considering the molecular formula and adjusting for the remaining two quaternary carbons, one oxygen and a hydrogen while at the same time accommodating the remaining DBE, the two subunits were united through the formation of a 1,1-3,3-tetramethyl-2,4-cyclohexaenone ring system. 

The planar structure confirmation and the absolute configuration for C-5″ was assigned to as *S*-based on the single-crystal X-ray diffraction (Mo Kα) data analysis (Figure 3), from which the Flack parameter of −0.1 allowed a confident configurational assignment to be made. 

**Figure 2 molecules-27-06592-f002:**
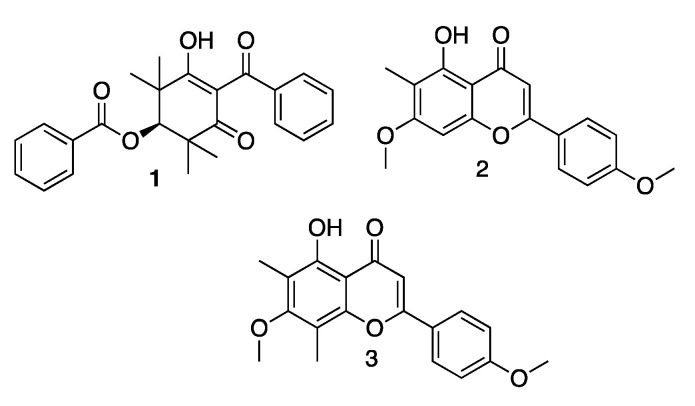
Structures of tristaenone A (**1**), 8-desmethyleucalyptin (**2**) and eucalyptin (**3**).

The known compounds were identified as 8-desmethyleucalyptin (**2**), eucalyptin (**3**) (Figure 2) by interpretation of its spectroscopic data (see Appendix A) and a close comparison with published data [13,14].

**Figure 3 molecules-27-06592-f003:**
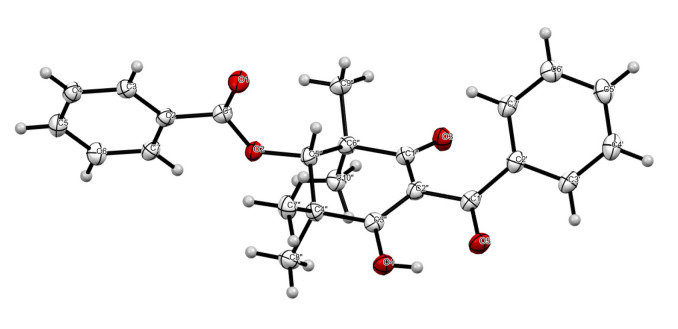
ORTEP diagram of **1**.

We investigated the anti-inflammatory activities of compounds **1**–**3** by evaluating the inhibition of NO production in lipopolysaccharides (LPS) plus interferon (IFN)-γ activated RAW 264.7 macrophages. All compounds were also evaluated for their cytotoxicity using the Alamar blue assay. All three compounds showed no cytotoxicity at concentrations up to 250 μM. Tristaenone A (**1**) and 8-desmethyleucalyptin (**2**) showed significant anti-inflammatory activity (inhibition of NO production) with IC_50_ values of 37.58 ± 2.45 μM and 16.21 ± 1.53 μM, respectively. However, the NO inhibitory effect of eucalyptin (**3**) was identified least with an IC_50_ of 138.47 ± 3.42 μM) (Table 2).

In addition, the effects of (**1**) on LPS and IFN-γ-induced expression of the pro-inflammatory cytokines TNF-α and IL-6 in macrophages were investigated. Tristaenone A (**1**) markedly decreased the production of TNF-α and IL-6 by showing IC_50_ values of 80.6 ± 5.82 μM and 125.65 ± 0.34 μM, respectively. Additional investigations of (**1**) on NF-κB translocation in LPS and IFN-γ-stimulated macrophages were performed to determine the mechanism underlying its anti-inflammatory effect. As expected, activation by the combination of LPS plus IFN-γ-induced NF-κB accumulation in the nucleus of RAW 264.7 cells. This had been represented in (Figure 4), where the pre-treatment of cells with different concentrations of (**1**) (31.85, 63.70 and 127.41 μM), significantly inhibited NF-κB nuclear translocation by 52.93 ± 14.14%, 59.38 ± 15.99% and 100 ± 14.14%.

## 3. Experimental Section 

### 3.1. General Experimental Procedures

UV spectra were recorded on a Shimadzu spectrophotometer model UV-2550. NMR spectra were recorded on a Bruker Avance 600 MHz spectrometer (Bruker Biospin GmbH, Rheinstetten, Germany) in the solvents indicated and referenced to residual ^1^H signals in deuterated solvents. Chiroptical measurements [α]_D_ were obtained on a Polax-D, ATAGO system polarimeter in a 100 × 2 mm cell at 25 °C. HRMS was carried out using a Waters Xevo Q-TOF mass spectrometer operating in the positive ESI mode.

### 3.2. Plant Material

The leaves of *T. laurina* were collected from the Australian Botanic Garden at Mount Annan (NSW, Australia) in July 2020. A voucher specimen (A1999-0386) has been deposited at the Australian Botanic Gardens, at Mount Annan, NSW, Australia.

### 3.3. Extraction and Bioactivity-Guided Isolation of Compounds ***1***–***3***

The fresh leaves of *T. laurina* (50 g) were crushed using a hand blender and extracted sequentially using organic solvents based on their polarity (n-hexane, dichloromethane (DCM), ethyl acetate (EtOAc), ethanol (EtOH), methanol (MeOH), and finally, water) using a Büchi-811 Soxhlet Extraction system. Immediately after the initial stages of sequential fractionation, each corresponding fraction was subjected to anti-inflammatory screening using the inhibition of NO in LPS plus IFN-γ-treated RAW 264.7 macrophages following Griess test (Appendix A). The most active extract (DCM) was solubilized in MeOH and was then later subjected to semi-preparative HPLC using an Agilent Zorbax C_18_ column (5 µm, 250 × 10 mm) column eluting at 2 mL/min from 10% MeCN/H_2_O to 100% MeCN (with a constant 0.01% FA modifier in the aqueous phase) over 20 mins and held for a further 40 min at 100% MeCN and then reconditioned back to 10% MeCN and maintained for an additional 10 mins, to yield (**1**) (t_R_ = 32.8, 20 mg), **2** (t_R_ = 32.0, 5.4mg) and **3** (t_R_ = 35.2, 5.5 mg). Compounds **1**–**3** were confirmed to be 99% pure based on LCMS and NMR analysis.

*Tristaenone A* (1): Colourless crystal [α]^25^_D_ −179.8 (c 0.01, MeOH); UV-Vis λ_max_ (MeOH) nm (log ε) 280 (5.34) and 233 (5.26); 1D and 2D NMR (600 MHz, CD_3_OD) data (see Table 1); HRESI (+)MS *m/z* 415.1521 [M + Na]^+^ (calcd for C_24_H_24_O_5_Na^+^; 415.1521).

### 3.4. X-ray Diffraction Analysis

Single crystal data was collected on the MX1 beamline at the Australian Synchrotron, using silicon double crystal monochromatic radiation (λ = 0.71073 Å) at 100 K [15]. The XDS software package [16] was used on site for data integration, processing and scaling. SADABS [17] was used to apply an empirical absorption correction. Shelxt [18] was applied to solve the structure by the intrinsic phasing method, and a suite of SHELX programs [18,19] were used for refinement, via the Olex2 graphical interface [20]. Crystallographic data of *tristaenone A* (CCDC 2172244) was deposited at the Cambridge Crystallographic Data Center. Additional crystallographic information is available in the Appendix A.

Crystal data for **1**: C_24_H_24_O_5_ (M = 392.43 g/mol); monoclinic, 0.1 × 0.1 × 0.01 mm^3^, space group *P*2_1,_
*V* = 991.0(4) Å^3^, *Z* = 2, *D*_c_ = 1.315 g/cm^3^, *F*(000) = 416.0, Mo Kα radiation, λ = 0.71073 Å, *T* = 100 K, μ = 0.092 mm^–1^; 2θ_range_ = 50.5°, 12,424 reflections collected, 3855 unique (*R*_int_ = 0.0453); final GooF = 1.066, R1 = 0.0323 [*I* > 2σ(*I*)], wR2 = 0.0825; absolute structure parameter = −0.1(3). 

### 3.5. Maintenance of RAW 264.7 Macrophages

Cells were grown in 75 cm^2^ flasks on DMEM containing 10% fetal bovine serum (FBS) that was supplemented with penicillin (100 μ/mL), streptomycin (100 μg/mL) and L-glutamine (2 mM). The cell line was maintained in 5% CO_2_ at 37 °C, with media being replaced every 3–4 days. Once cells had grown to confluence in the culture flask, they were removed using a rubber policeman, as opposed to using trypsin, which can remove membrane-bound receptors.

### 3.6. Pro-Inflammatory Activation of Cells

RAW 264.7 cells (1 × 10^6^ cells/mL) were seeded in 96 well plates (Corning^®^ Costar^®^, Sigma, Australia) overnight until confluency. When the cells were confluent, each compound was serially diluted from 100 μg/mL in two-fold dilution steps and co-incubated with cells for 1 h prior to the addition of 1 μg/mL LPS and 10 U/mL (1 unit = 0.1 ng/mL) IFN-γ. After activation, the cells were incubated for another 24 h at 37 °C. The supernatant was then collected for NO, TNF-α and IL-6 assays. The cells were subjected to cell viability measurements using the Alamar Blue assay. Non-activated cells (exposed to media only) were used as negative control and activated cells were positive control.

### 3.7. Determination of Nitrite by the Griess Assay

Nitric oxide was determined by the Griess reagent, as described in previous studies [21]. Griess reagent was freshly made up of equal volumes of 1% sulfanilamide in 5% phosphoric acid and 0.1% N-1-naphthylethylenediamine dihydrochloride in Milli-Q water. From each well, 50 µL of supernatant was transferred to a fresh 96-well plate and mixed with 50 µL of Griess reagent and measured at 540 nm in a POLARstar Omega microplate reader (BMG Labtech, Mornington, Australia). 

### 3.8. Determination of TNF-α and IL-6 by ELISA

The stored supernatants were analysed for TNF-α and IL-6 synthesis using a commercial ELISA kit (Peprotech, Queensland, Australia) according to the manufacturer’s instructions. The absorbance was measured at 410 nm [22]. The concentrations of TNF-α and IL-6 in the experimental samples were extrapolated from a standard curve.

### 3.9. Determination of NF-kB Translocation

NF-kB translocation was determined as described in previous studies [23]. RAW 264.7 cells were plated in an 8-well Nunc™ Lab-Tek™ II Chamber Slide™ System (Thermo Fisher Scientific, Sydney, Australia) at 20,000 cells/well. After 24 h, the cells were treated with compound **1** at 31.85, 63.70 and 127.41 μM in serum-free medium DMEM for 1 h prior to the stimulation of a combination of 1 μg/mL LPS and 10 U/mL (1 unit = 0.1 ng/mL) IFN-γ for 30 min. Cells were then washed with pre-chilled phosphate-buffered saline (PBS) buffer (Sigma-Aldrich, Melbourne, Australia) and fixed with 4% paraformaldehyde (Cell Signaling Technologies, MA, USA) for 30 min at room temperature. Triton X 100 (0.1%, Thermo Fisher Scientific, Australia) was added to permeabilize the cells for 20 min before being washed again three times with PBS and then blocked with 3% of bovine serum albumin (Bovogen Biologicals, Melbourne, Australia) for 1 h. The mouse anti-p65 NF-κB antibody (Santacruz, Australia) was co-incubated with cells overnight at 4 °C. Cells were rinsed (three times) with PBS again and incubated with the donkey anti-mouse IgG conjugated with Alexa Fluor 488 (green dye, Thermo Fisher Scientific, Sydney, Australia) for 1 h in the dark room at room temperature. After washing with PBS, the chambers were removed and the anti-fade mounting media with DAPI solution (blue colour, Sigma-Aldrich, Sydney, Australia) was added before capturing images with the Inverted Leica TCS SP5 laser scanning confocal microscope (School of Medicine, Western Sydney University, Australia). The fluorescent intensity was quantified and analysed using ImageJ following the previous study [24].

### 3.10. Determination of Cell Viability by the Alamar Blue Assay

After various treatments and the stimulation by LPS and IFN-γ overnight, 100 µL of Alamar Blue solution [10% Alamar Blue (Resazurin) in DMEM media] was added to cells and incubated at 37 °C for 2 h. The fluorescence intensity was measured with excitation at 530 nm and emission at 590 nm using a microplate reader. The results were expressed as a percentage of the intensity of that of control cells (non-activated cells).

### 3.11. Statistical Analysis

Data analysis was carried out using GraphPad Prism 9.3.1. Calculations were performed using MS-Excel version 16.61.1. IC_50_ values were obtained by using the sigmoidal dose–response function in GraphPad Prism. The results were expressed as mean ± standard deviation (SD).

## Figures and Tables

**Figure 1 molecules-27-06592-f001:**
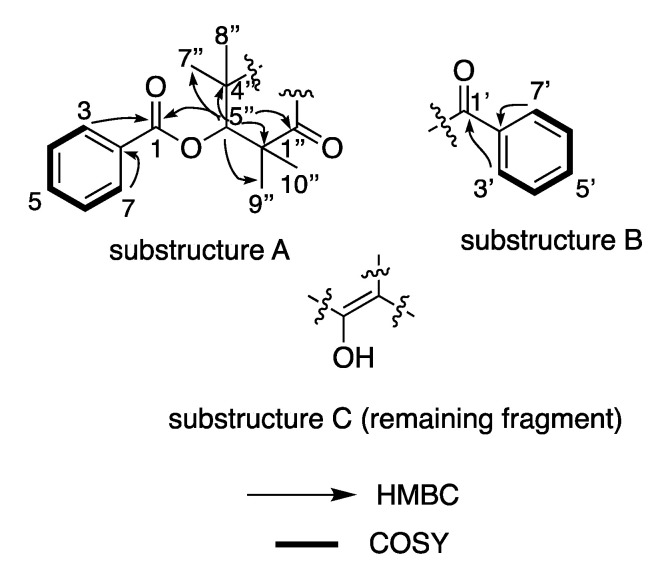
Diagnostic (600 MHz, CD_3_OD) HMBC and COSY correlations of **1**.

**Figure 4 molecules-27-06592-f004:**
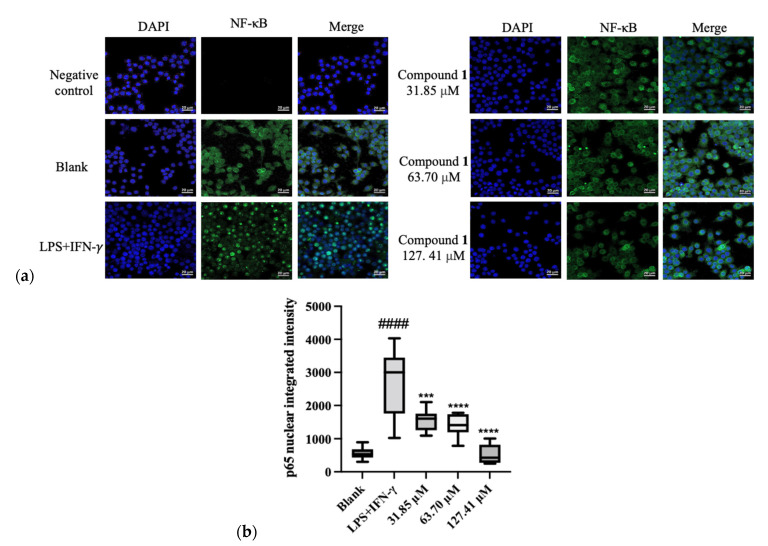
(**a**) RAW 264.7 cells were pre-treated with media (blank), compound 1 for 2 h, then stimulated with LPS and IFN-γ for 30 min. The cells were fixed and subjected to fluorescence staining with the mouse anti-p65 NF-κB antibody and Alexa Fluor 488 (green dye). The nuclei were stained with DAPI blue. Negative control refers to the RAW 264.7 cells with media, but not exposed to the mouse anti-p65 NF-κB antibody, and thus, not expressing the target antigen. The translocation of NF-κB (p65) was determined by using the immunofluorescence assay. Representative images were taken by confocal microscope with 40X magnification (scale bar = 20 μm). Blue: DAPI in the nucleus, green: NF-κB in the RAW 264.7 cells. (**b**) Quantification of % of nuclei positive p65 of staining in blank, LPS and IFN-γ-stimulated macrophages with and without various treatments.

**Table 1 molecules-27-06592-t001:** NMR data (600 MHz, CD_3_OD) for Tristaenone A (**1**).

Position	δ_H_ (*J* in Hz)	δ_C_ ^a^	COSY	HMBC
1		166.9		
2		132.9		
3/7	8.09, d (*7.8*)	130.4	4/6	1, 2, 3/7
4/6	7.54, dd (*7.8*, *7.6)*	129.6	3/7, 5	1, 2, 4/6
5	7.67, dd (*7.6, 7.6)*	134.5	4/6	4/6
1′		199.2		
2′		139.1		
3′/7′	7.60, d (*7.8*)	129.0	4″	1′, 2′
4″	7.45, dd (*7.8*, *7.4*)	128.8	3′/7′, 5′	2′, 4″
5′	7.54, dd (*7.4*, *7.4*)	132.9	4″	
1″		197.2		
2″		111.6		
3″		197.2		
4″/6″		46.2		
5″	5.51, s	80.2		1, 3″, 4″/6″, 7″/9″, 8″/10″
7″/9″	1.40, s	22.7		1, 3″, 4″/6″, 7″/9″, 8″/10″, 5″
8″/10″	1.40, s	25.8		1, 3″, 4″/6″, 7″/9″, 8″/10″, 5″

^a^ assignments supported by HSQC and HMBC experiments.

**Table 2 molecules-27-06592-t002:** Downregulation of LPS and IFN- γ-induced production of pro-inflammatory markers (NO and TNF-α) and cell viability of compounds (**1**–**3**) and the positive control curcumin.

Compounds	Inhibition of Nitric Oxide Production (µM)	Inhibition of TNF-α Production (µM)	Cell Viability (µM)
Tristaenone A (**1**)	37.58 ± 2.45	80.61	>250
8-desmethyleucalyptin (**2**)	16.21 ± 1.53	46.58	>250
Eucalyptin (**3**)	138.47 ± 3.42	>250	>250
Curcumin	12.6 ± 1.5	11.4 ± 1.3	29.5 ± 2.6

## Data Availability

Data supporting reported results can be accessed by contacting the corresponding author.

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
