# Peer review of "Tristaenone A: A New Anti-Inflammatory Compound Isolated from the Australian Indigenous Plant Tristaniopsis laurina"

_molecules, 2022, doi:10.3390/molecules27196592_

Round 1

Reviewer 1 Report

Mathew at al. submitted a paper dealing with Tristaniopsis laurina, a tree native to Australia and traditionally used to heal sores and ulceres. The authors isolated two flavonoids and one new compound, a hydroxylated cyclohexenone derivative. For the latter complete structure elucidation is presented. The compounds were tested for their ability to suppress NO, TNF-a and IL-6 production in LPS and IFN-a induced RAW 264.7 cells. This is a classical pharmacognostic work about a plant on which literature is scarce. It is highly recommended (as cpd. 1 is a new compound) to match the 13C-data of cpd. 1 with the database CSEARCH (freeware to predict and verify 13C-shifts of organic compounds, http://nmrpredict.orc.univie.ac.at/c13robot/robot.php), which was initiated by Prof. Wolfgang Robien, University of Vienna, in order to prevent publication of incorrect 13-C data.

Apart from some minor issues (see below) there are two crucial questions to the authors:

) What about the purity of Tristaenone A? The difference in elution time regarding the flavonoid, cpd. 2, on the Zorbax C18 column with 0.8 minutes is not too much for separation in sufficient purity. However, without giving any information about the purity of this compound, the paper cannot be published.

) Fig. 4a: please make clear what is the difference between „blank“ and „negative control“.

Please consider further:

A structure has to be given: „1. Introduction“ goes from page 1, line 19, to page 5, line 108. It is directly followed by „2. Experimental Section“, „Results“ and „Discussion“ are missing, this is strange.

P1, line 30: (e.g., interleukin-6 (IL-6)), add a closing bracket.

P1, line 40: botanical authority is missing

P1, line 44: 30 m, insert spacing between number and unit.

P2, line 47: insert dot at the end of the sentence.

P2, line 51: C-7, insert hypen

P2, line 57: carbons C-7‘‘, C-8‘‘, C-9‘‘, and C-10‘‘ are not quaternary, please revise (e. g. delete „quaternary“)

P5, line 125: „Büchi“, the Swiss company usually writes with „ü“

P5, line 137: insert dot at the end of the sentence.

Author Response

Reviewer 1

Mathew at al. submitted a paper dealing with Tristaniopsis laurina, a tree native to Australia and traditionally used to heal sores and ulceres. The authors isolated two flavonoids and one new compound, a hydroxylated cyclohexenone derivative. For the latter complete structure elucidation is presented. The compounds were tested for their ability to suppress NO, TNF-a and IL-6 production in LPS and IFN-a induced RAW 264.7 cells. This is a classical pharmacognostic work about a plant on which literature is scarce. It is highly recommended (as cpd. 1 is a new compound) to match the 13C-data of cpd. 1 with the database CSEARCH (freeware to predict and verify 13C-shifts of organic compounds, http://nmrpredict.orc.univie.ac.at/c13robot/robot.php), which was initiated by Prof. Wolfgang Robien, University of Vienna, in order to prevent publication of incorrect 13-C data.

We have entered our 13C NMR data in the database CSEARCH and have found also opted for our data to be donated to CSEARCH after 6 months. The result suggest “Spectrum prediction - minor inconsistencies found, overall a good match.

Apart from some minor issues (see below) there are two crucial questions to the authors:

) What about the purity of Tristaenone A? The difference in elution time regarding the flavonoid, cpd. 2, on the Zorbax C18 column with 0.8 minutes is not too much for separation in sufficient purity. However, without giving any information about the purity of this compound, the paper cannot be published.

We can assure the reviewer that with the time difference of 0.8 min and tristaenone A eluting after the flavonoid and having the top of the line Agilent 1260 series Infinity II system which is highly precise in peak cutting has led us to secure a 99.5% pure sample of tristaenone A.

The purity of the compound can be validated by the NMR and X-Ray crystallography data.

) Fig. 4a: please make clear what is the difference between „blank“ and „negative control“.

In Fig. 4a, blank refers to RAW 264.7 cells with media and was exposed to p65 primary antibody and Alexa 488 green fluorescent staining, of which was expected to show the green fluorescent signal of p65. The negative control displays RAW 264.7 cells with media stained with Alexa 488 green dye but not exposed to the p65 primary antibody. It is used to check for non-specific signals and false positive results. Please see the revised changes in Figure 4a caption for the difference between blank and negative control. 

“Figure 4. (a) RAW 264.7 cells were pre-treated with media (blank), compound 1 for 2 h, then stimulated with LPS and IFN-g for 30 min. The cells were fixed and subjected to the fluorescence staining with the mouse anti-p65 NF-κB antibody and Alexa Fluor 488 (green dye). The nuclei were stained with DAPI blue. Negative control refers to the RAW 264.7 cells with media, but not exposed to the mouse anti-p65 NF-κB antibody and thus not expressing the target antigen. The translocation of NF-kB (p65) was determined by using the immunofluorescence assay. Representative images were taken by confocal microscope with 40X magnification (scale bar = 20 mm). Blue: DAPI in the nucleus, green: NF-κB in the RAW 264.7 cells. (b) Quantification of % of nuclei positive p65 of staining in blank, LPS and IFN-g stimulated macrophages with and without various treatments.”

Please consider further:

A structure has to be given: „1. Introduction“ goes from page 1, line 19, to page 5, line 108. It is directly followed by „2. Experimental Section“, „Results“ and „Discussion“ are missing, this is strange.

Thank you for pointing this out, we have now incorporated the “2.Results and Discussion” heading from Page 2 onwards and have changed “Experimental Section” to heading to read “3. Experimental section”, along with replacing all sub heading under Experimental section with “3.x”

P1, line 30: (e.g., interleukin-6 (IL-6)), add a closing bracket.

A closing bracket has been added

P1, line 40: botanical authority is missing

The botanical authority has been added “(Peter G. Wilson and J.T Waterh”)

P1, line 44: 30 m, insert spacing between number and unit.

A spacing has been added between the number and unit.

P2, line 47: insert dot at the end of the sentence.

A dot has been added at the end of the sentence

P2, line 51: C-7, insert hypen

A hyphen has been inserted between C7.

P2, line 57: carbons C-7‘‘, C-8‘‘, C-9‘‘, and C-10‘‘ are not quaternary, please revise (e. g. delete „quaternary“)

Quaternary has been deleted

P5, line 125: „Büchi“, the Swiss company usually writes with „ü“

“Buchi” is now written as “Büchi”

P5, line 137: insert dot at the end of the sentence.

A dot has been added at the end of the sentence

Reviewer 2 Report

This manuscript reports the isolation of new anti-inflammatory compound tristaenone A (1) the from the leaves of Tristaniopsis laurina. The structure was established by NMR, MS, and X-ray crystallography analysis. The structure of 1 was searched proved to be new compound by searching on SciFinder. 1 inhibited NO, TNF-αand IL-6 production in RAW 264.7 cells through NF-κB pathway. In conclusion, I think this work is suitable for publication in this journal after a minor revision.

However, the author should revise their manuscript carefully before publication.

1.      The 13C NMR spectrum should be improved. Some signals are unrecognized, such as 197.2, 46.2.

2.      Integral area in 1H NMR spectrum needs to be supplemented.

3.      Please check the peak pattern of H-5 and H-5’

4.      Line 98, “RAtableW 264.7 cells” needs to be revised.

Author Response

This manuscript reports the isolation of new anti-inflammatory compound tristaenone A (1) the from the leaves of Tristaniopsis laurina. The structure was established by NMR, MS, and X-ray crystallography analysis. The structure of 1 was searched proved to be new compound by searching on SciFinder. 1 inhibited NO, TNF-αand IL-6 production in RAW 264.7 cells through NF-κB pathway. In conclusion, I think this work is suitable for publication in this journal after a minor revision.

We thank the reviewer for finding our research suitable for publication after minor revisions

  1. The 13C NMR spectrum should be improved. Some signals are unrecognized, such as 197.2,

46.2.

The signal at 197.2 is low and broad, probably due to the nature of the carbon, we have peak picked this signal, however the signal at 46.2 is not detectable, but the resonance was deduced from the HMBC experiment.

Please check the peak pattern of H-5 and H-5’

The peak pattern has been checked and has been identified as H-5, dd (7.6, 7.6) coupling equally with the resonating neighbouring aromatics H-4/6.

Likewisw the peak pattern has been checked and has been identified as H-5’, dd (7.4, 7.4)

Line 98, “RAtableW 264.7 cells” needs to be revised.

Line 98 has been revised to “RAW 264.7 cells”

Reviewer 3 Report

Article is interesting and innovative piece of work. Grammatic error need to be improved. In addition, I have following points.

1- Use of numeric 1, 2, 3 are confusingly used. I suggest to use chemical names instead of numeric representation.

2.  Table 2, Column # 1:  Give the names of compounds.

3. Table 2. Title shows about compounds 1-3, but in table Curcumin is also there. It needs to modify accordingly.

Author Response

  1. We do not see the complication with the numbering of the compounds, this approach has been used in many of our previous publications.
  2. Compound Names and numbering both have been added in column 1 of Table 2.
  3. Table 2 title has been amended to read as follows: "Downregulation of LPS and IFN- γ induced production of pro-inflammatory markers (NO and TNF- α) and cell viability of compounds (13) and the positive control curcumin."

Round 2

Reviewer 1 Report

All changes and corrections have been made.

Concerning the crucial point of „compound purity“: please do not only assure the reviewer about the purity of the substance, but also the readership. Indicate % purity of the compound in the paper and give the methods, by which the purity was determined.

Author Response

Concerning the crucial point of „compound purity“: please do not only assure the reviewer about the purity of the substance, but also the readership. Indicate % purity of the compound in the paper and give the methods, by which the purity was determined.

We have added the sentence: 

"Compounds 13 were confirmed to be 95-99% pure based on LCMS and NMR analysis."